# FAAH Inhibition Counteracts Neuroinflammation via Autophagy Recovery in AD Models

**DOI:** 10.3390/ijms252212044

**Published:** 2024-11-09

**Authors:** Federica Armeli, Roberto Coccurello, Giacomo Giacovazzo, Beatrice Mengoni, Ilaria Paoletti, Sergio Oddi, Mauro Maccarrone, Rita Businaro

**Affiliations:** 1Department of Medico-Surgical Sciences and Biotechnologies, Sapienza University of Rome, 04100 Latina, Italy; federica.armeli@uniroma1.it (F.A.); beatrice.mengoni@uniroma1.it (B.M.); 2European Brain Research Center, Santa Lucia Foundation IRCCS, 00143 Rome, Italy; roberto.coccurello@cnr.it (R.C.); giacomogiacovazzo@virgilio.it (G.G.); ilariapaoletti93@icloud.com (I.P.); soddi@unite.it (S.O.); mauro.maccarrone@univaq.it (M.M.); 3Institute for Complex Systems (ISC), National Research Council (C.N.R.), 00185 Rome, Italy; 4School of Veterinary Medicine, University of Teramo (UniTE), 64100 Teramo, Italy; 5Department of Biotechnological and Applied Clinical Sciences, University of L’Aquila, 67100 L’Aquila, Italy

**Keywords:** Alzheimer’s disease (AD), autophagy, BDNF, endocannabinoids, fatty acid amide hydrolase, microglia, mTOR, ATG7, neuroinflammation, URB597

## Abstract

Endocannabinoids have attracted great interest for their ability to counteract the neuroinflammation underlying Alzheimer’s disease (AD). Our study aimed at evaluating whether this activity was also due to a rebalance of autophagic mechanisms in cellular and animal models of AD. We supplied URB597, an inhibitor of Fatty-Acid Amide Hydrolase (FAAH), the degradation enzyme of anandamide, to microglial cultures treated with Aβ_25-35_, and to Tg2576 transgenic mice, thus increasing the endocannabinoid tone. The addition of URB597 did not alter cell viability and induced microglia polarization toward an anti-inflammatory phenotype, as shown by the modulation of pro- and anti-inflammatory cytokines, as well as M1 and M2 markers; moreover microglia, after URB597 treatment released higher levels of *Bdnf* and *Nrf2*, confirming the protective role underlying endocannabinoids increase, as shown by RT-PCR and immunofluorescence experiments. We assessed the number and area of amyloid plaques in animals administered with URB597 compared to untreated animals and the expression of autophagy key markers in the hippocampus and prefrontal cortex from both groups of mice, via immunohistochemistry and ELISA. After URB597 supply, we detected a reduction in the number and areas of amyloid plaques, as detected by Congo Red staining and a reshaping of microglia activation as shown by M1 and M2 markers’ modulation. URB597 administration restored autophagy in Tg2576 mice via an increase in BECN1 (Beclin1), ATG7 (Autophagy Related 7), LC3 (light chain 3) and SQSTM1/p62 (sequestrome 1) as well as via the activation of the ULK1 (Unc-51 Like Autophagy Activating Kinase 1) signaling pathway, suggesting that it targets mTOR/ULK1-dependent autophagy pathway. The potential of endocannabinoids to rebalance autophagy machinery may be considered as a new perspective for therapeutic intervention in AD.

## 1. Introduction

In recent years, much attention has been focused on endocannabinoids (eCBs) as promising signaling systems for developing therapeutic strategies against neurodegenerative diseases, including Alzheimer’s disease (AD) [1,2]. The eCB system is a lipid signaling network composed of endogenous lipid molecules and many proteins, enzymes and receptors. Anandamide (AEA) and 2-arachidonoylglycerol (2-AG) are the two majors lipid messengers binding, with a different affinity degree, to the G protein-coupled receptors cannabinoid type-1 (CB1) and type-2 (CB2) receptors, synthetized mainly by the N-acylphosphatidylethanolamine. Phospholipase D and diacylglycerol lipase are the main enzymes for the biosynthesis of AEA and 2-AG, while AEA and 2-AG degradation/inactivation is accomplished by hydrolyzing enzyme fatty acid amide hydrolase (FAAH) for AEA and monoacylglycerol lipase for 2-AG [3,4]. Since it is virtually involved in every physiological and pathophysiological process, the eCB system is recognized to play multiple pro-homeostatic functions [2], through the modulatory action exerted on neuroprotection, synaptic transmission and plasticity, as well as the resolution of inflammation [5,6]. Of particular interest for its therapeutic potential against AD is the hypothesis that the potentiation of eCB signaling via the inhibition of AEA or 2-AG catabolism might result in the reduction or prevention of neuroinflammation [6,7,8,9]. Neuroinflammation is a key hallmark of AD together with the well-described amyloid-beta (Aβ) aggregates forming extracellular plaques, and the neurofibrillary tangles originating from the accumulation of intracellular hyperphosphorylated tau protein [10]. However, despite the predominant view assigning to the Aβ cascade, the main responsibility for senile plaques and AD pathogenesis, every strategy aimed at blocking Aβ aggregates or depleting Aβ development has been revealed to be rather disappointing [11]. Since AD is a chronic condition, neuroinflammation can be a permanent state of neural injury characterized by chronic and sustained microglia overactivation characterized by a pro-inflammatory phenotype (so-called M1 phenotype) releasing pro-inflammatory mediators such as IL-1β (Interleukin 1 beta), TNF-α (Tumor necrosis factor alpha), IL-6 (Interleukin 6), and iNOS (Inducible nitric oxide synthase) and the impairment of the polarization towards an anti-inflammatory phenotype (so-called M2 phenotype) characterized by the release of anti-inflammatory factors such as IL-4 (Interleukin 4) and IL-10 (Interleukin 10), and neurotrophic factors (e.g., BDNF [brain-derived neurotrophic factor]) [12]. Accordingly, microglia overactivation and its accrual around Aβ plaques observed in AD-like mice have been related to a chronic neuroinflammatory condition, also because the Aβ peptide stimulates microglia to produce IL-1β, leading to an amplification of this process [13,14,15]. Nevertheless, microglia overactivation is described also in the absence of Aβ aggregates or the formation of Aβ plaques as observed in older adults with mild cognitive impairment [16,17], thus suggesting that early anti-inflammatory therapy may be of great benefict against microglia-mediated neuroinflammation [17]. Notably, microglia are tightly entangled with the eCB signaling system and not only microglial cells express CB1 and CB2 receptors, but they are also capable of producing and releasing eCBs [17,18,19,20,21,22]. The anti-inflammatory potential provided by the inhibition of eCB degradation is corroborated by the evidence that both the pharmacological inhibition and genetic inactivation of FAAH produce the downregulation of iNOS and cyclooxygenase-2, and reduce the expression of pro-inflammatory cytokines [23]. The genetic inactivation of the FAAH enzyme was shown to rescue synaptic plasticity in the hippocampus and increase phagocytosis and Aβ clearance in the 5xFAD mouse model of AD [24]. We have recently showed that FAAH inhibition via URB597 treatment is highly effective in promoting cytoskeleton reorganization in overactivated BV-2 microglia cells challenged with Aβ_25-35_ [9]. The powerful anti-inflammatory action induced by FAAH inhibition was also corroborated by the increase in phagocytic activity and the polarization of microglia towards the M2 phenotype [9]. Although defective autophagy is a crucial mechanism underlying neuroinflammation, oxidative stress, Aβ metabolism, and AD pathogenesis [25], there is still a paucity of knowledge about the relationship between autophagy malfunctioning, microglia polarization, and AD, as well as about the role of eCB signaling in the remodeling of microglia polarization. For this reason, we completed a set of experiments to address whether the increase in the eCB signaling via FAAH inhibition could produce a rescue of autophagy activity in BV-2 cells challenged with Aβ_25-35_ and, in parallel, in the Tg2576 AD-like transgenic mouse model [26]. In particular, we investigated the molecular effectors enabling autophagy restoration following FAAH inhibition, and the mechanisms providing anti-inflammatory responses and the improvement of AD pathology through the reshaping of microglia polarization and the modulation of selective components of eCB signaling.

## 2. Results

### 2.1. URB597 Treatment in BV-2 Cells 

We have previously demonstrated, via the radioligand anandamide, that URB597 inhibits the activity of FAAH in BV-2 cells [9]. URB597-treated cells showed no viability difference from control (CTRL), confirming the non-cytotoxicity of the drug. Combined Aβ + URB597 treatment increased the number of living cells compared to Aβ treatment and decreased the percentage of dead cells. As previously demonstrated by Grieco and collaborators (2009), we confirmed the anti-inflammatory activity achieved by URB597 treatment in the Aβ-treated BV-2 cells [9]. In particular, an increase in the iNOS levels (associated with M1 pro-inflammatory microglia phenotype) and a decrease in ARG-1 expression (associated with M2 anti-inflammatory microglia phenotype) were observed in the Aβ-treated BV-2 cells. Pre-treatment (4 h) with 5 μM URB597 decreased iNOS expression and increased ARG-1 expression thus further confirming the previous data [9]. 

Here, we evaluated the expression of the IL-6, IL-1β, *Tgf-β* and IL-10 mRNA levels by qPCR at 3 h and 24 h after Aβ challenge and URB597 treatment. The results (Figure 1) demonstrate that IL-1β and IL-6 mRNAs increased 3 h after Aβ treatment (*p* = 0.0092; *p* = 0.0098), while URB597 alone did not induce any expression of pro-inflammatory cytokines. On the other hand, a reduction in pro-inflammatory cytokines was observed at 3 h following treatment with URB597 as compared to the effects produced by the challenge with Aβ only (*p* = 0.0064; *p* = 0.0079). No significant changes in the mRNA’s expression of both IL-1β and IL-6 were evident at 24 h. On the other hand, the expression of IL-10 at 3 h increased significantly following Aβ challenge (*p* = 0.045), although the co-treatment with URB597 also increased IL-10 mRNA expression (*p* = 0.037). The *Tgf-*β mRNA expression did not show significant changes at 3 h after treatment. At 24 h, the expression of IL-10 and TGF-β increased significantly in the presence of URB597 (*p* = 0.0071; *p* = 0.0083) as well as in the combined URB597 and Aβ co-treatment (*p* = 0.00061; *p* = 0.038) as compared to Aβ alone. Hence, URB597 decreased the expression of anti-inflammatory cytokines in the short term, considering their early kinetics and, simultaneously, induced the secretion of anti-inflammatory cytokines 24 h after treatment.

The M2 phenotype plays a role in the resolution of inflammation via the release of anti-inflammatory cytokines, such as IL-4, IL-10, and TGFβ, as well as through the expression of antioxidant factors such as Nrf2 (Nuclear factor erythroid 2-related factor 2) [17]. The activation of Nrf2, following the increase in ROS (reactive oxygen species), regulates SQSTM1/p62 (sequestosome 1) gene expression and influences autophagy activity. This mechanism is deregulated in neurodegenerative diseases by the exacerbation of oxidative stress and neuroinflammation [27]. We evaluated, via RT-PCR, the *Nrf2* mRNA expression in URB597-treated or untreated BV-2 cells either in the absence or in the presence of Aβ. At 3 h after the Aβ challenge, *Nrf2* mRNA did not show any significant variation, whereas following URB597 treatment, we detected a slight significant increase in *Nrf2* (*p* = 0.002), and a strong significant increase with the combined treatment, i.e., URB597 + Aβ (*p* = 0.0000393). At 6 h after Aβ challenge, we found (Figure 2) a significant reduction in *Nrf2* mRNA expression as compared to the CTRL cells (*p* = 0.0009), while following URB597 treatment, *Nrf2* mRNA expression increased significantly also in the BV-2 cells challenged with Aβ (*p* = 0.0033; *p* = 0.000092). Thus, under stress neuroinflammatory conditions, URB597 promoted the increase of *Nrf2* mRNA expression.

To further explore the neuroprotective potential of URB597, we additionally evaluated the mRNA expression of *Bdnf.* promotes neuroprotection and modulates the synaptic interaction that is critical for cognition and memory, by increasing survival and inducing neuronal regeneration [28,29]. BDNF deficiency in AD contributes to neurodegeneration [30]. In particular, we assessed whether URB597 treatment was able to change BDNF mRNA expression in BV-2 cells either in the absence of or in the presence of Aβ challenge. As shown in Figure 3, there was a significant reduction in *Bdnf* mRNA expression in the presence of Aβ as compared to the untreated cells (CTRL) at both 1 h and 6 h after Aβ challenge (*p* = 0.000012; *p* = 0.00004). After 1 h, URB597 significantly increased BDNF mRNA expression in the presence of Aβ as compared to BV2 cells challenged with Aβ (*p* = 0.0002). At 6 h, *Bdnf* mRNA expression increased significantly in both URB597-treated cells (*p* = 0.000023), and in the combined treatment of Aβ + URB597 (*p* = 0.0143).

### 2.2. FAAH Inhibition in AD-like Mice

The expressions of microglia phenotypic markers such as iNOS and ARG-1 were evaluated via immunohistochemistry in the prefrontal cortex (PFC) and hippocampus of Tg2576 mice. We found an increase in iNOS-positive cells (PFC *p* = 0.026; hippocampus *p* = 0.0073) and a significant reduction in ARG-1-positive cells (PFC *p* = 0.0130; hippocampus *p* = 0.034), both in the PFC and the hippocampus of Tg2576 mice. URB597 administration decreased the iNOS expression in both the PFC (*p* = 0.000022) and the hippocampus (*p* = 0.000051). In particular, at the level of PFC, URB597 induced a lower expression of iNOS than that observed in the WT group (Figure 4A). Moreover, we disclosed a reduced expression of ARG-1 in Tg2576 mice compared to WT; URB administration produced an increase in ARG-1 in the Tg2576 animals compared to the vehicle-treated Tg2576 animals (PFC *p* = 0.000014; hippocampus *p* = 0.000036), but also compared to the URB597-treated WT animals (PFC *p* = 0.019; hippocampus *p* = 0.0016). Hence, URB597 administration elicited an increase in ARG-1-positive cells, both in the PFC and the hippocampus of Tg2576 mice (Figure 4B).

### 2.3. β-Amyloid Aggregation

The administration of URB597 resulted in a reduction in β-amyloid plaques. To assess the potential beneficial effect induced by FAAH inhibition on amyloid plaque accumulation, we analyzed both the number and area of plaques by using the Congo Red procedure. It was established that Tg2576 mice, particularly between 6 and 14 months of age, exhibited cognitive deficits and initial signs of brain atrophy. Around 14 months of age, this model began to develop extracellular Aβ accumulations [31]. Here, we administered Tg2576 mice at 14 months of age, revealing that early plaque accumulation was more evident in the midbrain region. This point agrees with one of the early features of AD, involving the degeneration of cholinergic fibers originating from the basal forebrain and innervating hippocampal and neocortical areas [32]. Figure 5 shows that in Tg2576 mice administered with URB597 there was less accumulation of amyloid plaques (number *p* = 0.000017; area *p* = 0.00091), as compared to untreated Tg2576 mice.

### 2.4. URB597 and Autophagy

We further assessed the expression of the major factors involved in the autophagy machinery in BV-2 cells treated with URB597, either in the presence or in the absence of Aβ. The panels in Figure 6A depict the mRNA expression levels of *Atg7* (autophagy-related 7), BECN1 (*Beclin 1*), *SQSTM1/p62*, and *Lc3*, at 6 h and 24 h post-treatment. ATG7 and BECN1 are essential for autophagy induction; LC3 is the main autophagosomal marker, and SQSTM1/p62 is an autophagy substrate that can be used as marker of autophagy flux. All four factors showed a significant reduction in mRNA expression in the presence of Aβ compared to BV-2 CTRL cells (BECN1 6 h *p* = 0.027, 24 h *p* = 0.000076; *Atg7* 6 h *p* = 0.000072, 24 h *p* = 0.00086; *Lc3* 6 h *p* = 0.00098, 24 h *p* = 0.00055; SQSTM1/p62 6 h *p* = 0.000015, 24 h *p* = 0.044) showing the suppressive effect exerted by Aβ on autophagy. Treatment with URB597 alone increased the mRNA expression of *Atg7* (*p* = 0.000077), BECN1 (*p* = 0.000081), *Lc3* (*p* = 0.0000023) and SQSTM1/p62 (*p* = 0.000060) at 24 h. Moreover, the combined treatment with Aβ and URB597 led to a significant elevation in mRNA expression compared to the CTRL cells (BECN1 6 h *p* = 0.038, 24 h *p* = 0.000059; *Atg7* 6 h *p* = 0.000082, 24 h *p* = 0.000046; *Lc3* 6 h *p* = 0.000098, 24 h *p* = 0.000065; SQSTM1/p62 6 h *p* = 0.000025, 24 h *p* = 0.0034). Importantly, these findings were consistent at both 6 h and 24 h, thus demonstrating that URB597 increased the mRNA expression of the main factors involved in autophagy flux. This suggests the involvement of URB597 and an FAAH blockade, in the restoration of impaired autophagy in AD. We next assessed the protein expression of BECN1 and ATG7 in Tg2576 or WT mice that underwent either URB597 or vehicle administration. A significant reduction in the expression of ATG7 and BECN1 was detected at the level of the hippocampus (*p* = 0.039; *p* = 0.031) and PFC (*p* = 0.027; 0.045) of Tg2576 mice compared to WT mice. These data confirm the alteration of proteins involved in the induction of autophagy. Following treatment with URB597, the Tg2576 mice exhibited a significant increase in the expression of BECN1 (PFC *p* = 0.039; hippocampus *p* = 0.000053) and ATG7 (PFC *p* = 0.0011; hippocampus *p* = 0.0013) in comparison to both Tg2576 and WT mice (Figure 6B). Moreover, we examined the expression of LC3 I-II and SQSTM1/p62 in the sagittal sections obtained from the PFC and hippocampus of Tg2576 mice administered with URB597, and compared the results with Tg2576 mice and WT mice. The results (Figure 6C) revealed a decreased amount of LC3 I-II in Tg2576 mice as compared to WT mice (PFC *p* = 0.029; hippocampus *p* = 0.033). URB597 administration produced a significant increase in the levels of LC3 I-II, thus restoring the values observed in both the PFC (*p* = 0.0041) and hippocampus (*p* = 0.036) in WT mice. In parallel, we observed a significant increase in SQSTM1/p62 expression in the hippocampus of Tg2576 mice, as compared to WT mice (*p* = 0.043) (Figure 6C), with a slight increase in PFC. After URB597 administration, the Tg2576 mice showed a significant reduction in SQSTM1/p62 expression as compared with the Tg2576 mice (PFC *p* = 0.0023; hippocampus 0.0091). 

### 2.5. FAAH Inhibition: LC3-II Expression in AD-like Mice

The precursor proLC3 was cleaved to form LC3-I and then modified via the conjugation to the phosphatidylethanolamine into the membrane-bound protein LC3-II, which has been characterized as an autophagosome marker in mammalian cells [33]. Our results confirmed the activity of autophagy in URB597-treated Tg2576 mice as shown by the significant increase via LC3-II expression in the hippocampus and PFC homogenates (Figure 7) compared to untreated (*p* = 0.026) and vehicle-treated Tg2576 mice (*p* = 0.0062).

### 2.6. mTOR and ULK1 Expression in AD-like Mice

The kinase mTOR (mammalian target of rapamycin) inhibits autophagy, and mTOR activity indeed appears deregulated in AD [34,35]. In our study, we assessed mTOR expression in sagittal sections from Tg2576 and WT mice treated with URB597, in comparison to mice from both of the groups treated with vehicle alone. The analysis revealed an increase in mTOR expression in both the PFC (*p* = 0.0073) and hippocampus (*p* = 0.028) from Tg2576 mice, as compared to the WT mice. By contrast, the Tg2576 mice administered with URB597 exhibited a significant reduction in mTOR expression in comparison to vehicle-treated Tg2576 mice, with a level of expression comparable to that observed in WT mice (PFC *p* = 0.00025; hippocampus *p* = 0.040) (Figure 8A). These findings suggest that URB597 treatment exerts a modulatory effect on mTOR expression, contributing to rebalancing the autophagy function. The observed reduction in mTOR expression supports the idea that FAAH inhibition may have a role as a positive regulator of autophagy. Next, we also examined the expression of ULK1 kinase (unc-51 like autophagy activating kinase 1), a factor involved in the activation of the autophagic process [36]. As shown in Figure 8B, our results indicated a significant reduction in the expression of ULK1 in both the PFC (*p* = 0.030) and the hippocampus (*p* = 0.019) from Tg2576 mice, as compared to WT mice. Furthermore, the expression levels of ULK1 were restored in Tg2576 mice administered with URB597 and were comparable to those observed in WT mice (PFC *p* = 0.00071; hippocampus *p* = 0.020). These findings provide evidence that URB597 treatment has a positive impact on ULK1 expression, potentially contributing to the activation of the autophagy in Tg2576 mice.

## 3. Discussion

Our study provides evidence that the blockade of FAAH activity in AD-like mice can offset neuroinflammation and promote autophagy restoration. Prior to the examination of the hypothesis that the inhibition of FAAH activity can reinstate autophagy in Tg2576 mice, we addressed the anti-inflammatory and antioxidant potential produced by URB597 treatment in BV-2 cells, following our previous demonstration that pre-treatment with URB597 reverts both the pro-inflammatory morphological phenotype and the migratory activity observed in Aβ_25-35_-incubated BV-2 cells [9]. We assessed the ability of URB597 treatment to reduce pro-inflammatory cytokines such as IL-1β, TNF-α, and IL-6, and to increase the release of anti-inflammatory cytokines such as TGF-β and IL-10 in BV-2 cells, either in the absence or in the presence of Aβ_25-35_. Interestingly, the IL-10 and TGF-β mRNA levels were increased after co-treatment with Aβ_25-35_ and URB597. Similarly, we evaluated the impact of the Aβ_25-35_-elicited M1-like inflammatory phenotype in microglial cells and URB597-induced anti-inflammatory potential by assessing the decrease in iNOS as well as the switching towards an anti-inflammatory M2-like phenotype mirrored by the increase in ARG-1 expression. We next assessed the possibility that URB597-treated BV-2 microglia might activate specific intracellular anti-inflammatory, antioxidant, and neuroprotective mechanisms such as Nrf2, which is involved in the maintenance of redox homeostasis [37,38]. FAAH inhibition produced an increase in Nrf2 mRNA expression, especially after the Aβ_25-35_ challenge of BV-2 microglia. However, since the ubiquitous expression of Nrf2 in the brain, to evaluate the defensive mechanisms elicited by the inhibition of FAAH activity, we also assessed the mRNA expression of *Bdnf*, whose levels have been described to change according to the intensity of oxidative damage [39,40]. Interestingly, URB597 alone increased *Bdnf* mRNA expression in BV-2 microglial cells while Aβ_25-35_ and URB597 co-treatment reinstated the BDNF expression previously reduced by Aβ_25-35_ treatment. Thus, not only did URB597 treatment stimulate the *Bdnf* expression in microglia, but it also abolished the suppressive effect produced by Aβ_25-35_ challenge. On this ground, we ascertained whether these markers of microglial M1/M2 polarization were changed by in vivo subchronic URB597 administration in Tg2576 mice. The blockade of FAAH activity produced a marked decrease in iNOS positive cells in both the hippocampus and PFC, which was increased in AD-like animals. Moreover, Tg2576 animals showed a reduction in ARG-1, an index of the M2 phenotype, in both the PFC and hippocampus, but URB597 administration reverted such decrease by increasing ARG-1 positive cells. The polarization towards the M2 phenotype may have increased microglia phagocytic activity, possibly reducing amyloid plaques aggregation [41], as shown by the evaluation of the plaque area, and induced the release of anti-inflammatory cytokines. Aβ deposition was lower in the midbrain of 14-month-old Tg2576 mice treated with URB597, a brain area in which alterations of cholinergic synaptic markers have been described even before the onset of amyloid plaque deposition [42]. Being aberrant in AD [43], autophagy malfunctioning exacerbates the accumulation of misfolded proteins (e.g., Aβ). Several key factors of the autophagy machinery such as BECN1, *Atg7*, *Lc3 and SQSTM1/p62* that resulted in downregulated BV-2 microglia upon Aβ challenge increased after URB597 treatment. In parallel, URB597 administration increased BECN1 and ATG7 in the hippocampus and the PFC of Tg2576 AD-like mice. The increase in eCB signaling via the inhibition of FAAH activity also produced an increase in LC3-II, which is a marker of autophagosomes, further demonstrating the stimulatory effect induced on autophagy activity. Consistently, the reduced levels of SQSTM1/p62 expression corroborated the induction of autophagy after URB597 administration. Notably, mTOR expression was increased in AD-like mice while URB597 administration produced its inhibition, which was an effect of key importance for the increase in Aβ clearance [44]. Autophagy activity was tightly regulated along the pathway of AMP-activated protein kinase (AMPK/mTOR/ULK1/2), which has been shown to be important for the improvement of AD-associated neuropathology [36,45]. ULK1 expression was reduced in both the hippocampus and PFC from TG2576 mice, while the inhibition of FAAH activity increased ULK1 activation. Among the other mechanisms underlying the effects of FAAH inhibition on autophagy activation, it was important to consider those attributable to other FAAH substrates such as palmitoylethanolamide and N-oleoylethanolamide, whose elevation can stimulate brain peroxisome proliferator-activated receptors (PPARs)–α receptors [46,47]. The pharmacological activation of PPAR-α receptors stimulated autophagy in the microglia as well as in the cells expressing human APP, also decreasing Aβ accumulation and attenuating cognitive impairment in AD-like mice [48]. Palmitoylethanolamide can exert a marked anti-inflammatory action by reducing the expression of pro-inflammatory factors such as iNOS [49], thus contributing to explaining the decrease in the iNOS positive cells that we found in the hippocampus and PFC after URB597 administration. It should be kept in mind that autophagy can be a “double-edged weapon” in which both defective or excessive activity can be highly detrimental and lead either to the insufficient cytoprotection and degradation of damaged proteins or to cell death, in both cases compromising cell homeostasis. For instance, URB597 has been shown to reduce the abnormal autophagy induced by chronic cerebral hypoperfusion [50]. On the other hand, chronic cerebral hypoperfusion is also responsible for NLRP3 inflammasome activation and defective autophagy, which can be recovered by URB597 administration by preventing ROS accumulation and microglia overactivation [51]. Although controversial [52], either the selective activation of CB1 or the administration of the mixed CB1/CB2 agonist Sativex^®^ can stimulate autophagy in intestinal epithelium [53], the brain cortex, and the hippocampus [54]. Moreover, Δ9-THC administration has been shown to increase the expression of LC3-II, an effect reverted by the CB1 blockade [55]. Remarkably, since CB1 receptors have also been described to be located on the external membrane of brain mitochondria [56], the modulation of CB1 receptor signaling has been shown to increase mitophagy (i.e., mitochondrial autophagy) in hippocampal neurons [57]. Altogether, our study provides evidence that subchronic URB597 administration counteracts iNOS expression and that increasing brain eCB signaling and, in particular, AEA signaling, can be a valuable strategy to reprogram a microglia phenotype whose hyperactivation has been identified as a pathogenetic factor underlying iNOS overactivation, in both AD-like models and brains from AD patients [58,59,60].

These results were obtained by in vitro and in vivo studies and were confirmed by analyzing monocytes/macrophages from AD patients; as a matter of fact, the monocytic CB2 and FAAH levels significantly correlated with clinical scores [61]. In addition, the FAAH substrate AEA was reduced in the mid-frontal and temporal cortex of AD patients where it was inversely correlated with Aβ42 content [62], paralleling the findings obtained in animals.

## 4. Limitations

Although our study was accurately planned, the overall interpretation may have some limitations.

The aggregation of misfolded amyloid-structured proteins leads to the deposition of toxic insoluble cellular as well as extracellular aggregates, which are the pathological hallmarks of AD. A limitation of our study is the absence of the determination of intracellular hyperphosphorylated tau tangles which may have disclosed the role of the increased endocannabinoid tone in autophagy recovery and the impact of autophagy function on tau pathology.

We administered URB597 to Tg2576 AD-like mice and observed the restoration of autophagy through the increased expression of Beclin1, ATG7, LC3, and p62, as well as the activation of the ULK1 signaling pathway. However, cognitive performance was not evaluated in our study. Validating the disease phenotype in the Tg2576 model through behavioral tests would have added the opportunity to link the observed mitigation of iNOS expression and the modulation of the microglial phenotype, via the enhancement of eCB signaling, specifically AEA signaling, with cognitive performance. Such an approach would have strengthened the translational relevance of our findings. Moreover, considering the multifaceted nature of Alzheimer’s disease and its impact on different brain areas, investigating additional regions could have yielded insights into the impact of URB597 addressing the potentiation of eCB signaling.

## 5. Materials and Methods

### 5.1. In Vivo Experimental Design

Transgenic Tg2576 mice expressing high levels of mutated human APP (amyloid-beta precursor protein, Swedish K670N/M671L mutation) and showing severe cognitive decline, the brain aggregation of amyloid plaques, neuroinflammation and synaptic loss were used [63]. The Tg2576 mice were heterozygous for the APP-K670N/M671L transgene and were obtained by crossing hemizygous males (Tg2576-F0) with female C57BL/6J/SJL-F0 hybrid mice. The latter were obtained by crossing SJL males with WT C57BL/6J females. The mice were group-housed (3–4 mice/cage) with temperature (22–23 °C) and humidity (60 ± 5%) control under a 12:12 h light/dark cycle, with food and water freely available throughout the study. All the procedures were performed in accordance with European and Italian national law (DLGs n.26 of 04/03/2014, European Communities Council Directive 2010/63/UE) about the use of animals for research (Italian Ministry of Health, authorization n. 421/2019-PR). Male, fourteen-month-old Tg2576 and WT mice were used. Genotyping was performed to validate the overexpression of the human mutant APP DNA sequence via polymerase chain reaction (PCR). The mice were genotyped between 20 and 25 days of age via tail biopsy analysis. The tails were collected to extract genomic DNA, according to standard techniques; 10 ng of DNA per sample was amplified by PCR. Genotyping was carried out using the following primers: (FW 5′-CTG ACC ACT CGA CCA GGT TCT GGG T-3′, REV 5′-GTG GAT AAC CCC TCC CCC AGC CTA GAC CA-3′; by Sigma Aldrich), as previously specified [64]. AD-like Tg2576 and WT male mice were intraperitoneally (i.p.) administered with the FAAH inhibitor URB597 (Selleck Chemicals, S2631, Houston, TX, USA) either at 10 mg/kg/day (URB 10) or 0 mg/kg/day (URB 0) in a single dose for a total of 8 consecutive administrations (8 days). The dose of URB597 administered via the i.p. route was determined on the basis of a previous comparison between different routes of administration [65]. URB597 was dissolved in a vehicle composed 90% of sterile 0.9% NaCl, 5% PEG-400 (polyethylene glycol), 5% Tween-80 (Sigma-Aldrich, Milano, Italy) [65]. The animals were sacrificed on day 8 of treatment, 2 h after their last injection. The animals were randomly assigned to four experimental groups, as follows: (1) WT URB 0 mg/kg; (2) WT URB 10 mg/kg; (3) Tg2565 URB 0 mg/kg; Tg2576 URB 10 mg/kg. Randomization was achieved via the online tool for random number generators (https://www.randomizer.org/), accessed on 5 September 2022. Normality was determined using the Shapiro–Wilk normality test.

### 5.2. In Vitro Analysis

We have already demonstrated the ability of URB597 to inhibit the activity of FAAH in BV-2 cells challenged by Aβ_25-35_ (Sigma-Aldrich, A4559, St. Louis, MO, USA) [9]. We prepared the stock solution of URB597 [3-(3-carbamoylphenyl) phenyl] N-cyclohexylcarbamate as previously described [9]. Aβ_25-35_ was dissolved in sterilized distilled water, at a concentration of 1 mM. To induce the aggregation, the solution was incubated in a 37 °C water bath for 7 days, and then stored at −20 °C for preservation.

### 5.3. Cell Cultures and Treatments

A mouse microglia cell line (BV-2), kindly provided by Dr. Mangino (Sapienza University of Rome, Italy), was seeded in Dulbecco’s Modified Eagle’s Medium High Glucose (Sigma Aldrich, D5671-500ML), 10% of Fetal Bovine Serum (Sigma Aldrich, F7524, St. Louis, MO, USA), 1% L-glutamine (Aurogene, AU-X0550, Rome, Italy), 1% penicillin–streptomycin (Aurogene, AUL0022, Rome, Italy), 1% non-essential amino acids (Aurogene, AU-X0557, Rome, Italy) and 1% sodium pyruvate (Sigma Aldrich, S8636, St. Louis, MO, USA), at 37 °C in a humidified atmosphere with 5% CO2. Immortalized cell lines and BV-2 cells derived from the neonatal brain cells of inbred C57BL/6 mice infected with the J2 retrovirus that carries the v-raf/v-myc oncogene, have been widely utilized as an in vitro culture model [66]. Cells were plated at an appropriate density according to each experimental setting and, after 24 h, treated with Aβ_25-35_ 30 μM in the presence or absence of URB597 5 μM. The cells were pre-treated with URB597 for 4 h before adding the Aβ_25-35_.

### 5.4. Real-Time Quantitative PCR Analysis

Total RNA was extracted from the control and treated BV-2 cells using the miRNeasy micro kit (Qiagen, 1071023, Hilden, Germany) and quantified with NanoDrop One/OneC (Thermo Fisher Scientific, Waltham, MA, USA). The cDNA was generated using the High-Capacity cDNA Reverse Transcription Kit (Applied Biosystems, 4368814, Waltham, MA, USA). cDNA was mixed with 0.3 μL of both forward and reverse primers and using the Power SYBR^®^ Green PCR Master Mix (Applied Biosystem, 4367659, Waltham, MA, USA) to a final reaction volume of 30 μL. Quantitative real-time PCR (qPCR) was performed for each sample in triplicate on an Applied Biosystems 7900HT Fast Real-Time PCR System (Applied Biosystem, Waltham, MA, USA) using the SDS2.1.1 program (Applied Biosystem, Foster City, CA, USA). The thermal cycling parameters for primer optimization were as follows: reactions were performed with a hot start step of 48 °C for 30 min followed first by 40 amplification cycles (denaturation occurred at 95 °C for 15 s and annealing/extending occurred at 60 °C for 1 min). A melt curve was generated using Applied Biosystems 7900HT Fast Real-Time PCR System through the program SDS2.1.1.

Primers for real-time PCR amplification were designed with UCSC GENOME BROWSER (http://genome.cse.ucsc.edu/; (accessed on 1 November 2022) University of California, Santa Cruz). The sequences of the primer pairs were matched via BLASTn to the genome sequence to identify the positions of the primers relative to the exons (Table 1). The comparative threshold cycle (CT) method was used to analyze the real-time PCR data. The target quantity, normalized with respect to the endogenous reference of the 18S rDNA primers (ΔCT) and relative to the calibrant of the untreated control (ΔΔCT), was calculated with Equation 2^−ΔΔCT^.

### 5.5. Immunohistochemistry

Animals allotted for the immunohistochemical analysis underwent intracardial perfusion with phosphate-buffered saline (PBS) at a concentration of 0.1 mol/liter and a pH of 7.4, using a 30 mL syringe. Next, perfusion was prolonged with a 4% paraformaldehyde solution in PBS. Following perfusion, the brains were extracted from the skulls and post-fixed in a 4% paraformaldehyde solution for 16 h overnight at 4 °C. The brain specimens were then transitioned through a 30% sucrose solution in 0.1 mol/liter phosphate buffer for 48–72 h until equilibrated. The brains were frozen via immersion in cold isopentane for 3 min, sealed in vials, and stored at −80 °C until needed. The frozen sections (10 µm thickness) were obtained with a cryostat, placed on BDH slides (Leica, Milan, Italy), and stored at −20 °C. At the time of analysis, the sections were thawed, and three washes in PBS were conducted. Endogenous peroxidases were blocked in 3% hydrogen peroxide in methanol for 12 min. After three additional washes in PBS, non-specific binding sites were blocked with blocking serum (Scytek Ultratek HRP Polyvalent AFN600, Logan, UT, USA) for 12 min. Following three washes in PBS, the sections were incubated with primary antibodies, and diluted in PBS and 0.1% bovine serum albumin, overnight at 4 °C (Anti-iNOS 1:100 (Novus Biological, NB300-605, Milan, Italy); Anti-ARG-1 1:100 (Immunological Sciences, AB-84248, Rome, Italy); Anti-LC3I-II 1:100 (MBL, PM036MS, Schaumburg, IL, USA); Anti-BECN-1 1:100 (Immunological Sciences, AB-82599, Rome, Italy); Anti-mTOR 1:100 (Immunological Sciences, AB-84433, Rome, Italy); Anti-ULK1 1:100 (Immunological Sciences, AB-84102, Rome, Italy); Anti-ATG7 1:100 (Immunological Sciences, AB-83978, Rome, Italy); SQSTM1/p62 1:100 (Santa Cruz Biotechnology, sc-28359, Dallas, TX, USA).

Subsequently, the sections underwent three 5 min washes in PBS and were then incubated with biotinylated secondary antibodies specific to either a mouse or rabbit origin for 12 min (Scytek, AFN600, Logan, UT, USA). Excess antibodies were removed via 5 min washes in PBS, and the reaction was visualized using the complex Ultratek HRP Polyvalent (Scytek, AFN600, Logan, UT, USA) conjugated to a peroxidase enzyme. This enzyme reacted with the DAB chromogen (Scytek DAB Substrate Kit, ACH500, Logan, UT, USA), forming an insoluble colored precipitate visible under a light microscope. The nuclei were counterstained with hematoxylin (Sigma Aldrich, MHS16, St. Louis, MO, USA) for 5 min, followed by a 10 min rinse under running water to remove excess stain. The sections were allowed to air-dry at room temperature, covered with a glass coverslip, and mounted with an aqueous solution (Scytek, AML500, Logan, UT, USA). The percentage of positive cells was quantified using Nikon’s universal software platform, NIS-Elements (AR, version 4.30.02., Nikon, Tokyo, Japan). The following primary antibody was dissolved at a concentration of 1:100 in PBS.

### 5.6. Congo Red Method

We examined the presence of amyloid plaques via the Congo Red method. The sections were defrosted, and the nuclei were highlighted with hematoxylin for 5 min; staining under running water was performed for 10 min. After that, the sections were covered with Congo Red 1% aqueous solution (Electron Microscopy Sciences, 26090-25, Hatfield, PA, USA) for 1 h and washed with PBS; staining under running water was performed for 5 min. The sections were left to dry at RT and were closed with glass coverslip and aqueous mounting. The area (μm^2^) and number of plaques was assessed by Nikon’s universal software platform, NIS-Elements (Nikon, Tokyo, Japan).

### 5.7. ELISA Assay

To distinguish the LC3-II form from the LC3-I form, we took advantage of an ELISA assay. Homogenates of the cortex and hippocampus from untreated and vehicle/URB597-treated Tg2576 and WT mice were prepared and analyzed according to the Autophagy LC3-II Quantitation ELISA Kit (Cell Biolabs, CBA-5116, San Diego, CA, USA). The absorbance was read following the kit directions with a Varioskan LUX multimode microplate reader (Thermo Fisher Scientific, Waltham, MA, USA).

### 5.8. Statistics

Data were expressed as the mean values ± standard deviations (SD) or the mean values ± SEM and, in the case of in vitro assays, from at least three independent measurements. Statistical analyses were performed using unpaired Student *t* test (GraphPad Software Inc., Prism 8.0.1, San Diego, CA, USA). All the results were considered statistically significant with *p* < 0.05.

For the in vivo assays, the data were expressed as the mean values ± standard deviations (SD) or mean values ± SEM from at least three independent experiments (i.e., for the cell study). Statistical analyses were performed using a two-tailed unpaired Student *t* test (by GraphPad Software Inc., San Diego, CA, USA) to compare the data sets that passed the normality verification. All the results were considered statistically significant with *p* < 0.05. As the exclusion criteria of animals from the experiment, we established a priori that would have allowed any welfare issue to determine the exclusion from the experimental procedure. Thus, the animals’ welfare has been monitored after the first day of URB597 administration, particularly the animals’ conditions (body weight, wounds, fur) and the expression of behaviors (motility, grooming, sleep/wake cycle, feeding) in their home cage. None of the experimental units (i.e., mice) in the four planned experimental groups were excluded from the analysis. During the implementation of the experimental procedure, no strategies to control confounders were applied, because common potential confounders (e.g., order of treatments and measurements, animal location) were not present in the experimental design. The sample size was estimated on the basis of previous experiments and specific power analysis (assuming 80% power at a significance level of 0.05, by using G*Power 3.1 software). As reported in the figure legends, the sample size was always *N* = 6 for each experimental group, except for determination of LC-3 protein (via ELISA assay) in which, on the basis of previous sample size estimation, *N* = 3.

## 6. Conclusions

The “proof-of-concept” of the anti-inflammatory activity relies on the ability of URB597 treatment to limit the expression of pro-inflammatory cytokines such as IL-1β, TNF-α and IL-6, as well as to increase the release of anti-inflammatory cytokines. 

Because URB597 inhibits FAAH, it increases the levels of eCBs such as AEA, which is known to have neuroprotective and anti-inflammatory activity. This discovery opens new perspectives for therapies based on the modulation of the eCBs system, an emerging area of research against neurodegenerative diseases. URB597’s ability to reduce the expression of pro-inflammatory cytokines and increase the expression of the anti-inflammatory ones, suggests that it may be exploited as a drug to reduce brain inflammation and slow the progression of disease. The activation of the *Nrf2* pathway and the increased *Bdnf* expression indicate that URB597 may promote the survival and function of neurons in humans. This is critical to counteract the neuronal loss and cognitive decline characteristic of neurodegenerative diseases. Increasing *Bdnf* may support synaptic plasticity, and improve memory and cognitive abilities. In addition, the fact that URB597 can restore the normal functioning of the mTOR/ULK1 pathway for autophagy suggests that it could help reduce the toxicity associated with protein accumulation, thereby improving neuronal health and potentially delaying disease onset or its worsening. Although these findings are promising, they were obtained in animal models; therefore, the next challenge will be to test the safety and efficacy of URB597 in humans. Future clinical studies will need to determine the optimal dose, any side effects, and the drug’s efficacy on a large scale. If the results are positive, URB597 may represent a new class of drugs for the treatment of neurodegenerative diseases. Humans and mice have substantial differences in drug metabolism, which may affect the bioavailability, distribution, metabolism, and elimination of URB597. In mice, the administration of URB597 could lead to different concentrations in the brain and blood compared to humans. Therefore, it is possible that effective and safe doses in mice may not be directly transferable to humans. Pharmacokinetic studies in humans will be needed to establish appropriate dosages. Dose conversion between mice and humans may not be linear and will require the detailed evaluation of half-life time, hepatic metabolism, and interaction with other drugs. AEA potentiation through FAAH inhibition might have different effects on mood and behavior in humans than in mice. While mice do not experience the same psychoactive effects, in humans, the balance between the neuroprotective benefits and any adverse effects on the psychic sphere, such as anxiety or altered mood, must be carefully evaluated. In conclusion, if the preclinical results obtained with URB597 are confirmed in humans, this molecule could offer a novel therapeutic approach for the treatment of AD, not only by alleviating inflammation and improving neuroprotection, but also by directly reducing amyloid deposits and improving cellular elimination mechanisms via autophagy.

## Figures and Tables

**Figure 1 ijms-25-12044-f001:**
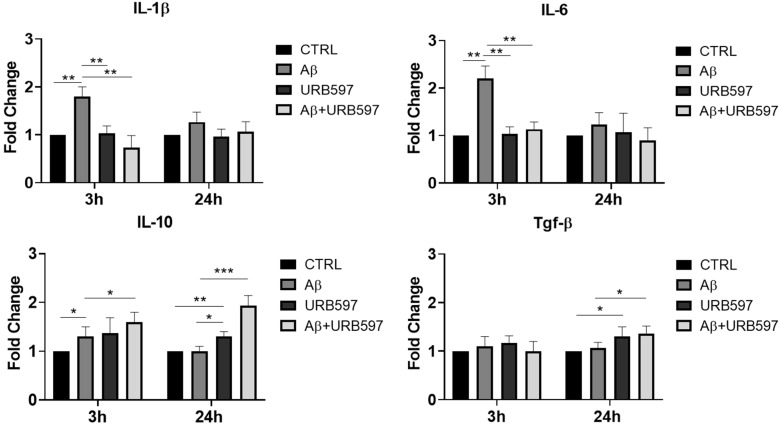
The mRNA expression of different inflammatory cytokines. mRNA of IL-1 β, IL-6, *Tgf-β*, and IL-10, monitored by qPCR and normalized to 18 S ribosome subunit. Data are shown as mean ± SD from three independent experiments performed in triplicate. Expression profiles were determined using the 2^−ΔΔCT^ method. * *p* < 0.05, ** *p* < 0.01, *** *p* < 0.001.

**Figure 2 ijms-25-12044-f002:**
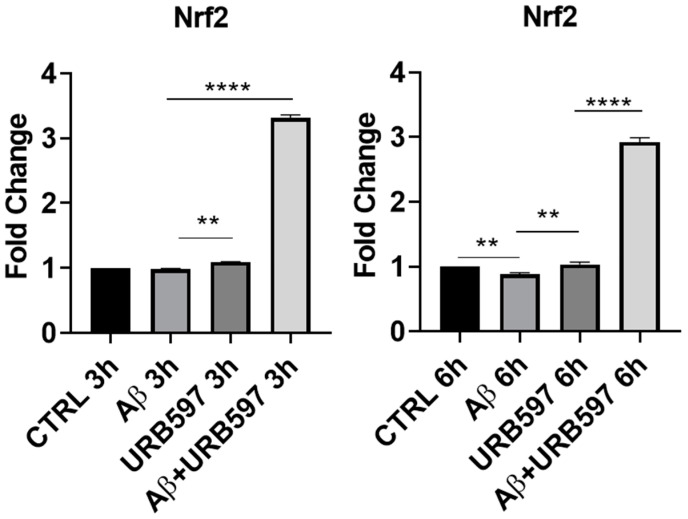
mRNA expression of *Nrf2* was evaluated by qRT-PCR at 3 h and 6 h. Data are shown as mean ± SD from three independent experiments performed in triplicate. Expression profiles were determined using the 2^−ΔΔCT^ method. ** *p* < 0.01, **** *p* < 0.0001.

**Figure 3 ijms-25-12044-f003:**
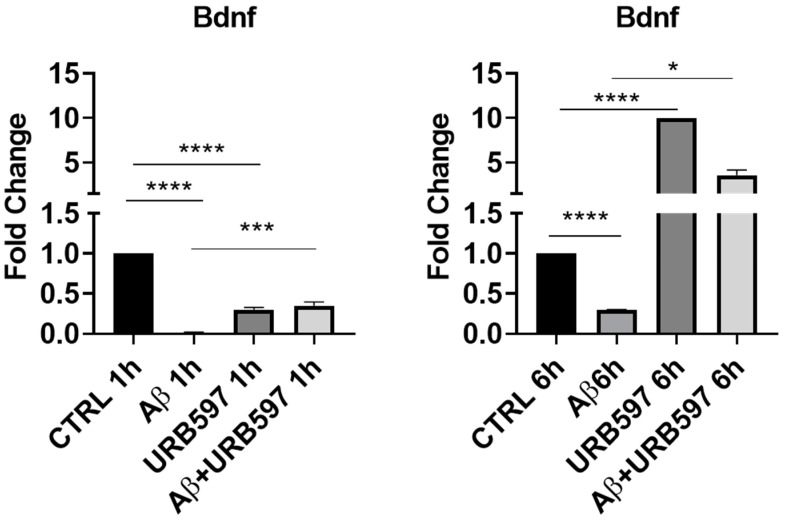
mRNA expression of *Bdnf* was evaluated by qRT-PCR at 1 h and 6 h. Data are shown as mean ± SD from three independent experiments performed in triplicate. Expression profiles were determined using the 2^−ΔΔCT^ method. * *p* < 0.05, *** *p* < 0.001, **** *p* < 0.0001.

**Figure 4 ijms-25-12044-f004:**
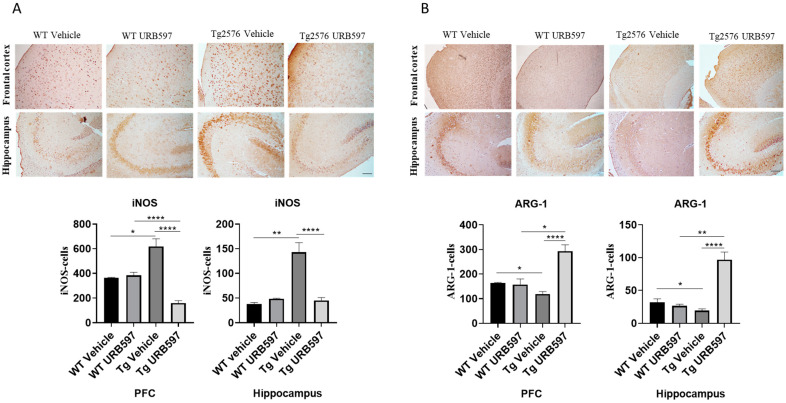
Immunohistochemical analysis (Ematoxillin and DAB chromogen) in sagittal sections of PFC and hippocampus. (**A**) shows decreased iNOS marking in Tg2576 mice treated with URB597; (**B**) shows increased ARG-1 marking in Tg2576 mice treated with URB597; these results were confirmed by quantification shown in graphs. Analysis was performed by considering 4 10× magnification fields for both hippocampus and cortex, and values were expressed as mean ± SD of total positive cells expressed in both brain regions from 6 independent experiments for each experimental group using unpaired Student *t* test. Significance levels were denoted as follows: * *p* < 0.05, ** *p* < 0.01, **** *p* < 0.0001.

**Figure 5 ijms-25-12044-f005:**
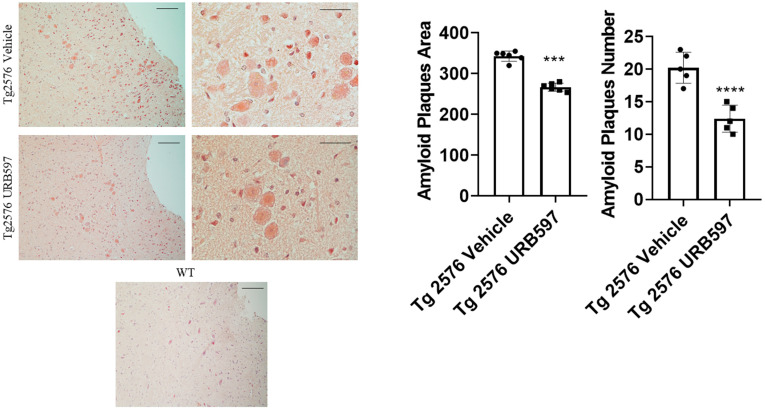
Congo Red analysis in sagittal sections including either vehicle or URB597-treated Tg2576 mice. Images show amyloid plaques. Analysis was performed by considering 4 magnification fields; area was expressed in μm^2^ and values were expressed as mean ± SD of total positive cells expressed in both brain regions from 6 independent experiments for each experimental group, using unpaired Student *t* test. Significance levels were marked as follows: *** *p* < 0.001, and **** *p* < 0.0001. Magnification 10× and 40×.

**Figure 6 ijms-25-12044-f006:**
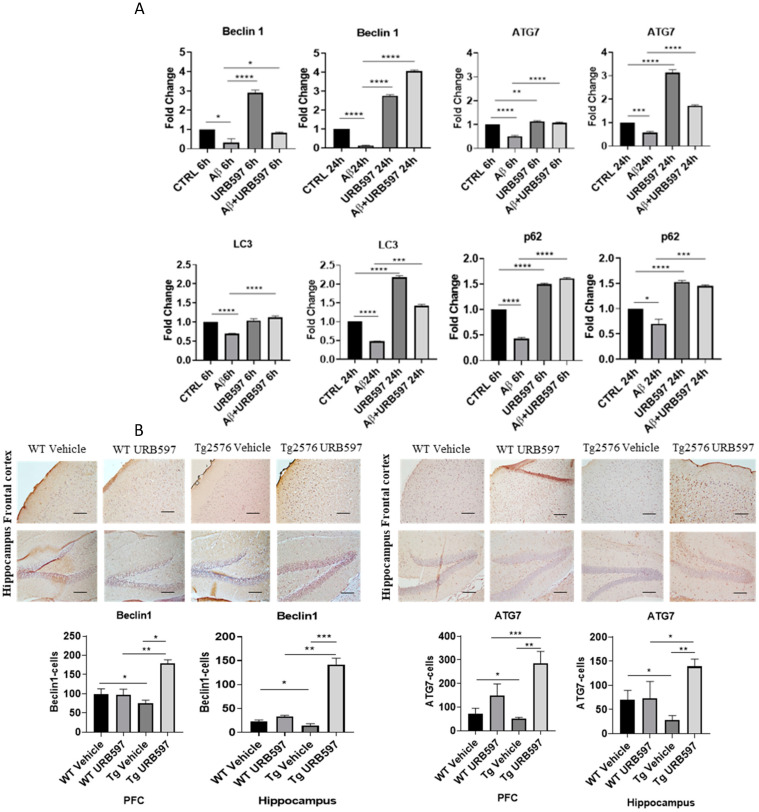
Analysis of mRNA expression of factors involved in the autophagic process. (**A**) mRNA expressions were evaluated by qRT-PCR at 6 h and 24 h of *Atg7*, BECN1, *Lc3*, and *SQSTM1/p62*. Data are shown as the mean ± SD from three independent experiments performed in triplicate. Expression profiles were determined using the 2^−ΔΔCT^ method. Significance levels were denoted as follows: * *p* < 0.05, ** *p* < 0.01, *** *p* < 0.001, **** *p* < 0.0001. (**B**) Immunohistochemical analysis (Ematoxillin and DAB chromogen) in sagittal sections of vehicle-treated WT and Tg2576 experimental groups and WT and Tg2576 treated with URB597. The images show ATG7- and BECN1-positive cells in PFC and hippocampus. Analysis was performed by considering 4 10× magnification fields for both hippocampus and PFC, and the values were expressed as the mean ± SD of the total positive cells expressed in both of the brain regions from 6 independent experiments for each experimental group using an unpaired Student *t* test. Significance levels are indicated as follows: * *p* < 0.05, ** *p* < 0.01, *** *p* < 0.001. (**C**) Immunohistochemical analysis (Ematoxillin and DAB chromogen) in sagittal sections of vehicle-treated WT and Tg2576 groups and WT and Tg2576 administered with URB597. The images show the LC3I-II- and SQSTM1/p62-positive cells in the PFC and hippocampus. Analysis was performed by considering four 10× magnification fields from the hippocampus and PFC, and the values were expressed as the mean ± SD of the total positive cells expressed in both of the brain regions from 6 independent experiments for each experimental group, using an unpaired Student *t* test. Significance levels are indicated as follows: * *p* < 0.05, ** *p* < 0.01.

**Figure 7 ijms-25-12044-f007:**
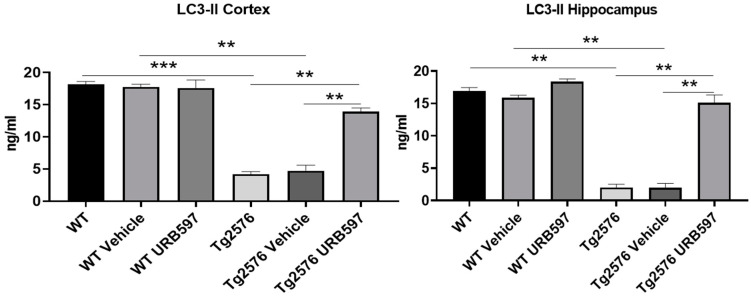
LC3-II ELISA assay on homogenates of PFC and hippocampus from untreated and vehicle/URB597-treated Tg2576 and WT mice. Data are shown as mean ± SD from three independent experiments performed in duplicate using unpaired Student *t* test. Significance levels were denoted as follows: ** *p* < 0.01, *** *p* < 0.001.

**Figure 8 ijms-25-12044-f008:**
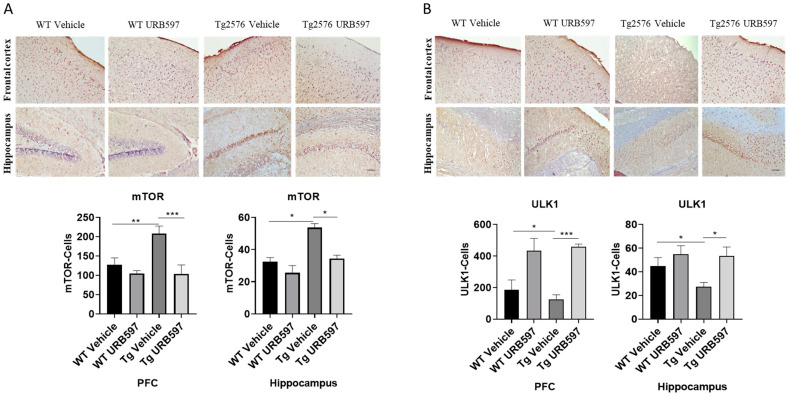
Immunohistochemical analysis (Ematoxillin and DAB chromogen) in sagittal sections of vehicle-treated WT and Tg2576 experimental groups and WT and Tg2576 treated with URB597. (**A**) The images show mTOR positive cells in the PFC and the hippocampus. Analysis was performed by considering 4 10× magnification fields for both hippocampus and PFC, and the values were expressed as the mean ± SD of the total positive cells expressed in both the brain regions from 6 independent experiments for each experimental group using an unpaired Student *t* test. Significance levels were denoted as follows: * *p* < 0.05, ** *p* < 0.01, *** *p* < 0.001. (**B**) The images show ULK1-positive cells in the PFC and the hippocampus. Analysis was performed by considering 4 10× magnification fields for both the hippocampus and the PFC, and the values were expressed as mean ± SD of the total positive cells expressed in both of the brain regions from 6 independent experiments for each experimental group using an unpaired Student *t* test. Significance levels were denoted as follows: * *p* < 0.05, *** *p* < 0.001.

**Table 1 ijms-25-12044-t001:** List of primers couples generated for qPCR.

GENE	Forward Primer (5′–3′)	Reverse Primer (5′–3′)	Accession Numbers
*mIL-1β*	GAAATGCCACCTTTTGACAGTG	TGGATGCTCTCATCAGGACAG	NM_008361.4
*mIL-6*	CGGAGAGGAGACTTCACAGAGGA	TTTCCACGATTTCCCAGAGAACA	NM_001314054.1
*mTgf-β*	CTCCCGTGGCTTCTAGTGC	GCCTTAGTTTGGACAGGATCTG	NM_011577.2
*mIL-10*	GCCCTTTGCTATGGTGTCCTTTC	TCCCTGGTTTCTCTTCCCAAGAC	NM_010548.2
*mR18s*	AAATCAGTTATGGTTCCTTTGGTC	GCTCTAGAATTACCACAGTTATCCAA	M27358
*mLc3*	TTCTTCCTCCTGGTGAATGG	GTCTCCTGCGAGGCATAAAC	NM_026160
*mBeclin1*	CAGCCTCTGAAACTGGACACGA	CTCTCCTGAGTTAGCCTCTTCC	NM_019584
*mNrf2*	TCTGAGCCAGGACTACGACG	GAGGTGGTGGTGGTGTCTCTGC	NM_010902
*mp62*	CCTTGCCCTACAGCTGAGTC	CCACACTCTCCCCCACATTC	NM_001290769
*mAtg7*	CAATGAGATCTGGGAAGCCATAA	AGGTCAAGAGCAGAAACTTGTTGA	NM_001253717
*mBdnf*	GTGTGACAGTATTAGCGAGTGG	GCAGCCTTCCTTGGTGTAAC	NM_007540

## Data Availability

The raw data supporting the conclusions of this article will be made available by the authors on request.

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
