# Peer review of "FAAH Inhibition Counteracts Neuroinflammation via Autophagy Recovery in AD Models"

_ijms, 2024, doi:10.3390/ijms252212044_

Round 1
Reviewer 1 Report
Comments and Suggestions for Authors
As the authors report, ‘one process altered in AD includes autophagy, which normally removes damaged or worn cellular organelles, and misfolded proteins detected in AD neurons’.
Aiming at rebalancing the autophagy mechanism, by cell culture AD-like mice, the authors investigated whether inhibition of FAAH activity by administration of URB597 may restore autophagy.
In this reviewer’s opinion, the manuscript, likely to be of interest, needs to be completely reorganized as follows:
1. The guidelines for article titles report as follows: single, concise sentence, ideally fewer than 10 words, summarizing the content. In contrast, the title of the manuscript as well as the titles of the Result section report the conclusions in a single sentence. With respect to a very long Result section, the short length of the Discussion and even more of the Conclusion unbalances the manuscript. The lines of the manuscript must be numbered and the paragraphs too.
2. The aim of the manuscript and the design of the study is not clear to the reader. The abstract starts introducing the role of Endocannabinoids with respect to inflammation and their potential role in reducing autophagy. General considerations have not place in an Abstract and must be eliminated. The remaining lines of the abstract are confusing. The authors must explain with much more clarity and schematically the performed measurements together with the adopted method.
Particularly
a) BV-2 cells viability, pro-inflammatory and anti-inflammatory cytokines, mRNA expression of M1 and M2 microglia markers and BDNF mRNA expression were analyzed’.
The adopted methods and the main results obtained must be shortly reported.
b) To assess the potential beneficial eZect induced by FAAH inhibition on amyloid plaque accumulation, we analyzed both number and area of plaques by using the Congo red procedure.
The same as before.
The abstract must contain the main overall results instead of the generic ending statement.
3. Materials and Methods section
a)Materials must not only be mentioned a description of their use and the
quantity used must be here reported. Based on those reported in the text, the reader must be able to reproduce your experiments.
b) Information about cell lines must be added, their origin by kindness is not
enough, nor plays so much importance for the reader.
The in vivo experimental design could be better placed at the beginning of the section.
Statistic methods must be reported in a unique paragraph.
4. A separate paragraph reporting the Limitations of the study is required.
5. The study is conducted on mice cells. Considering the totally diZerent metabolism and physiology what is the expectation on man?
6. Neuroinflammation is one of the crucial variables the authors refer to. It is also mentioned among the keywords. To investigate this point, other parameters are suggested to be investigated: specific biomarkers are available and/or, much better, EPR method is indicated.
7. Bar graphics must be enlarged. In this version are very diZicult and sometimes impossible to read variables etc.
8. Finally, the manuscript must be carefully revised by a native English speaker.
Comments on the Quality of English LanguageExtensive editing required
Reviewer 2 Report
Comments and Suggestions for Authors
General comment:
The article explores the potential of endocannabinoids to counteract neuroinflammation associated with Alzheimer's disease (AD), with a particular focus on restoring autophagy, the process by which cells remove damaged organelles and misfolded proteins.
The authors' goal is to evaluate whether inhibition of the FAAH enzyme by administration of URB597 can remodel the polarization of microglia and restore autophagy, thus contributing to the improvement of AD symptoms.
Methods: Using BV-2 microglia cells and Tg2576 mice. Analyzing cell viability, proinflammatory and anti-inflammatory cytokines, and expression of microglia markers M1 and M2. Evaluation of the expression of key autophagy markers by immunohistochemistry and ELISA. Analysis of amyloid plaque accumulation using the Congo red procedure.
Main results: FAAH inhibition remodeled microglial activation to an anti-inflammatory phenotype. Administration of URB597 restored autophagy in Tg2576 mice. Increased autophagy markers and activation of the ULK1 signaling pathway. Possibility of restoring the mTOR/ULK1-dependent autophagy pathway.
Strong points
1. Investigating FAAH inhibition and the effects of URB597 on neuroinflammation and autophagy is an innovative field with therapeutic potential. The results have important implications for the development of new therapeutic interventions in the treatment of AD.
2. The manuscript is clear, original and relevant to the field and well structured.
3. The cited references are relevant.
4. The study is designed correctly and technically sound
5. The manuscript is written in a logical and easy-to-understand manner.
6. The conclusions are consistent with the evidence and arguments presented.
7. The article is of interest to the scientific community.
8. The statements and conclusions are coherent.
Weaknesses.
1) Authors should also support their results on the basis of similar results in the literature, in order to reinforce their significance.
Minor corrections:
1. Figure 1 – missing – needs to be added
2. I believe that the "Real-time quantitative PCR analysis" does not contain enough data to be reproduced. I also suggest grouping the former used in a table to improve and make them easier to observe.
3. I suggest writing the name of the genes used in the study in accordance with the rules of the nomenclature in force: The names of human genes are written in large letters and italics, followed by a number or a symbol (e.g. BDNF). The names of genes in mice are written in small letters and italics (e.g. Bdnf).
4. Page 11 – in vivo – should be written in italics
5. "Statistics" - in vitro - should be written in italics
6. "Materials" - Aβ25-35 - Aβ25-35
7. After the number of figures, a period must be added.
8. Between figure 2 and 3 their description is too much space, it needs to be reduced.
9. Due to the multitude of terms and methods used, it is recommended to include an abstract graphic to facilitate understanding.
